# Attempt to Combine Physicochemical Data with Thermal Remote Sensing to Determine the Extent of Water Mixing between River and Lake

**Remigiusz Tritt [1], Adam Młynarczyk [2,\*] and Jędrzej Proch [3]**

1   Chair of Tourism and Recreation, Faculty of Geographic and Geological Sciences,
    Adam Mickiewicz University, Krygowskiego 10, 61-680 Poznań, Poland
2   Environmental Remote Sensing and Soil Science Research Unit, Faculty of Geographic and Geological
    Sciences, Adam Mickiewicz University, Wieniawskiego 1, 61-712 Poznań, Poland
3   Department of Analytical Chemistry, Faculty of Chemistry, Adam Mickiewicz University,
    Uniwersytetu Poznańskiego 8, 61-614 Poznań, Poland
\*   Correspondence: adam.mlynarczyk@amu.edu.pl

**Abstract:** The mixing of river and lake waters is important for the functioning of a reservoir, especially in the case of shallow polymictic reservoirs such as Lake Swarzędzkie. The extent of this mixing depends largely on the river flow rate. In lakes, which rivers with low flow values flow through, it should be expected that the flow currents only reach the narrow zone adjacent to the mouth of the river to the lake. The water of rivers generally has different chemical compositions and physicochemical parameters in relation to lake water. Therefore, to determine the range of the river in the lake and characterize the water mixing, measurements of temperature, electrolytic conductivity, and the concentrations of selected chemical elements were made in the estuary zone and at other points located on the lake and on the river near the tributary. In addition, the values and directions of horizontal currents were determined, and thermal photos were taken from a low-altitude ceiling.

**Keywords:** flow currents; Swarzędzkie Lake; water mixing; thermal images

## 1. Introduction

The horizontal movement of lake water is the result of the wind's impact on the water's surface and, in the case of flow-through reservoirs, also the effect of the inflow of river water. The scale of the watercourse's impact on the lake depends on the size of the reservoir, flow intensity, or lake morphometry. The presence of submerged macrophytes in the reservoir is also important and can significantly affect water fluxes throughout the lake [1]. Therefore, the flow may only reach the narrow zone of the river mouth or be along the entire length volume of the lake. The extent of mixing is the spatial extent of the zone of mixing between river and lake waters, where parameters (such as temperature) differ from the rest of the reservoir. The influence of the river flow on the physicochemical properties of lake water differs in periods of stagnation and homothermic periods [2]. The water of the rivers always flows in the layer of lake water that has similar temperature and density. Incoming water with a lower density than lake water will flow on the surface of the reservoir, and water with a higher density will fall to a layer of the same density or fall to the bottom of the reservoir [3]. Depending on the season and the river water temperature relative to that of lake water, the mixing of river and lake waters may also vary, occurring either at the surface or at the bottom of the reservoir [4,5]. In the case of strongly flowing lakes, when the temperature of lake and river water differs, it is possible to determine the impact zone by simultaneously measuring the temperature at many points of the reservoir [6].

Another way of determining the influence of the river is the thermal imaging method, which allows the determination of temperature differences simultaneously for a significant

area of the reservoir. The disadvantage of this method is that it only determines the temperature of the surface water layer. An additional problem when using UAVs is wind above a dozen m/s, which makes it impossible for small aircraft to fly, and the mixing of lake waters increases with the wind speed.

Another method is to study the isotope composition in river and lake water [7]. Sorokovikova [8], while studying the mixing of the water of the Selenga River with the water of Lake Baikal, apart from determining the chemical composition and transparency of the water, also determined the spatial differentiation of the quantitative and qualitative composition of phytoplankton. Cimatoribus et al. [9] determined the movement of river water through the lake using Lagrangian numerical transport modelling, tracking water particles flowing from the river into the lake.

Remote sensing in the thermal range is based on the measurement of electromagnetic radiation emitted from the surface of an object and can be applied to the measurement of water surface temperature. There are many factors affecting the remote measurement of this surface temperature, such as air temperature and humidity, wind speed, distance from objects, recording time, and the sensor used for the measurement [10–12]. Using an Unmanned Aerial Vehicle (UAV) and thermal camera, it is possible to study water surface temperatures [13–17]. The most commonly used wavelength range is 8–14 μm [18] at resolutions of 640 × 512 pixels [19], and the final products are thermal orthophotos [20–24]. In particular, the use of thermal imaging in the study of surface waters and their monitoring relate to identifying the place of occurrence of warmer waters, including sewage, discharged into rivers and lakes [14] and estimation of heat emission in the area of geothermal lakes [11], imaging of thermal variability of the lake surface [21].

In addition to the methods mentioned for determining the extent of mixing between river and lake waters, which include measurements of physicochemical parameters (primarily temperature), element and chemical concentrations, and thermal measurements, attempts can be made to determine this phenomenon using modern techniques for measuring water movement, such as the acoustic Doppler method. This method, based on Euler's theory, using the Acoustic Doppler Current Profiler (ADCP) device, gives precise results for measuring water velocity [25,26]. In lake conditions, moored, floating [27], and bottom-mounted ADCPs are mainly used [25,28].

The aim of this study was to determine the extent of the mixing of river waters with waters of a shallow polymictic lake using the example of Swarzędzkie Lake and a small river Cybina. To describe this phenomenon in detail, the study used all the above-mentioned research methods, i.e., measurements of physical parameters (temperature, electrolytic conductivity), measurements of chemical elements concentrations, measurements of water dynamics, and thermal images. This also made it possible to compare these methods and to determine which of them is the most beneficial for such a shallow lake and river characterised by low flows.

## 2. Materials and Methods

### 2.1. Study Area

Lake Swarzędzkie is in the urban-rural commune of Swarzędz in Poznań County, in the central part of the Wielkopolska Voivodeship (Figure 1). It is an intensively developing commune. The increasing number of inhabitants and noticeable economic and industrial development affect the natural environment, which is strongly anthropogenic transformed [29]. Swarzędz Lake is almost directly adjacent to the residential development of the city of Swarzędz to the east and south. Such a location may further intensify the negative impact on the lake's trophic status [30].

Swarzędzkie Lake is located to the south and west of the Poznan phase of the North-Polish glaciation [31]. The relief around the lake is characterised by small elevations. Along the NE-SW axis, the deeply incised Cybina valley is clearly marked here. The slopes of the valley edge reach up to 10%, and the width is a maximum of 500–800 m [32]. The geomorphological map of the Wielkopolska Lowland [33] shows that the valley at the level

of Swarzędzkie Lake meets a flat moraine plateau, in the north-east, it meets an undulating moraine plateau, and in the west, it meets a sand plain.

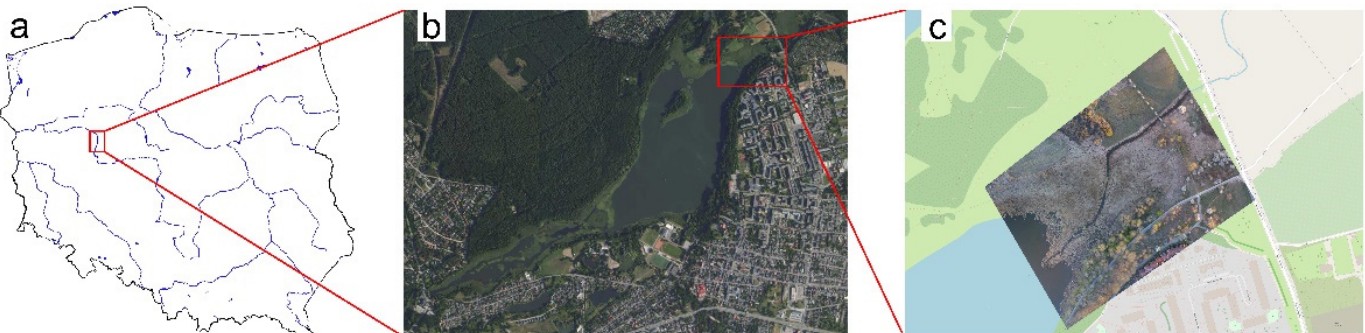

**Figure 1.** Location of the research area in Poland (**a**), Lake Swarzędzkie (**b**), thermal imagery area (**c**).

According to the division of the Wielkopolska Lowlands into climatic regions, Lake Swarzędzkie is located in the XV region-Central Wielkopolska [34]. In this region, wind directions are only slightly modified by the relief. Winds from the west and south-west are predominant, while those from the north and north-east are the least frequent. Most often, these winds are very weak (up to 2 m/s) and weak (2–5 m/s). Stronger winds are observed only occasionally [35]. The average annual air temperature in this area is 8.0 °C, and the annual precipitation varies between 500 and 600 mm [35].

Lake Swarzędzkie is a flow-through lake. The river Cybina, 41 km long, flows through them [36]. Its catchment area of 195.5 km$^2$, according to the hydrographic division of Poland, is determined by the topographic division of the third order [37]. The bad condition of Cybina's water is undoubtedly influenced by the discharge of municipal and industrial sewage, pollutants of agricultural origin, and periodic water discharges from fish ponds. The Cybina Valley, within which Lake Swarzędzkie is located, is a Natura 2000 protected area. Lake Swarzędzkie is a shallow post-glacial reservoir with a maximum depth of 7.2 m and an average depth of 2.6 m [38,39]. In 2017, the maximum depth of the lake was set at 7.5 m [40]. The area of the reservoir is 93.7 ha [36,39]. The lake does not develop a hypolimnion, and only about 15% of the bottom surface is within the range of the metallymnion [36], It is a polymictic lake. The shape of the reservoir stretches from the northeast to the southwest. There are two islands-one large, wooded, and the other smaller, covered with reeds. Lake Swarzędzkie is a eutrophic reservoir, mainly because of strong anthropopressure. The wastewater management in the commune was put in order, and in 2011, the lake was sustainably recultivated. Three methods were used simultaneously: phosphorus inactivation, bottom water oxygenation, and biomanipulation [39]. Despite these treatments, the general condition of the water is still described as bad.

The Cybina river, flowing through Lake Swarzędzkie, is the right tributary of the Warta. It has its springs near the villages of Siedleczek and Nekielka [36]. Fluctuations in the states of water for this river show one rising period and one low flow, characteristic of the snow and rain supply regime. Maximum states are observed in the spring period, as a rule, in March. Low flows generally last from June to November (Figure 2). The area in question is in the zone of the lowest outflow in Poland, which results from the deficit of rainfall and low water retention capacity of the area [41]. Mean annual flow values have trended downwards in recent years, and the mean annual flow during the measurement period was only 0.15 m$^3 \cdot$s$^{-1}$ with one period of low and one period of high flows [42]. In 2017 and 2018, according to the report of the Chief Inspectorate for Environmental Protection, the chemical status of Cybina's water was described as below good, and the general condition of the water was described as poor [43]. It is associated with the discharge of municipal and industrial wastewater, pollution of agricultural origin and periodic discharge of water from ponds.

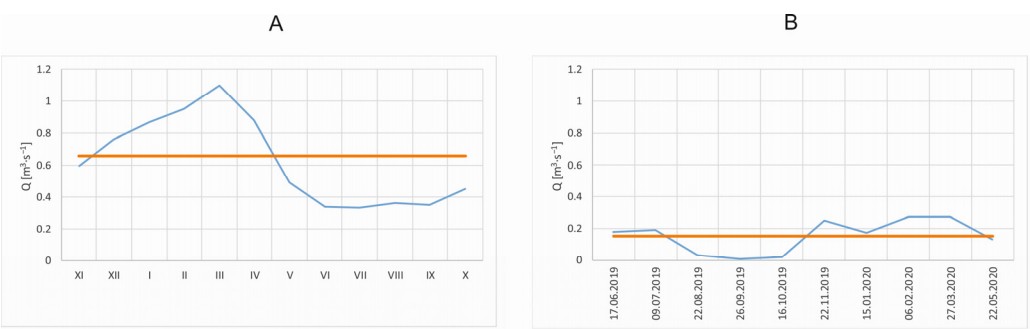

**Figure 2.** Average monthly-blue line and average annual–orange line flow for Cybina, Antoninek profile in 1976–1980 (**A**) [41]; daily average flow from June 2019 to May 2020-blue line and average of these measurements (**B**) [42].

*2.2. Methods*

The results used in this paper are a fragment of a wider project concerning the study of water circulation in Swarzędzkie Lake. The basic criterion for the selection of measurement points was the morphometry and bathymetry of the reservoir. The idea was, among other things, to create measurement profiles on both sides of the island. Points at the inflow and outflow were selected so that changes in water dynamics could be determined in the context of the influence of the stream. The latest bathymetric plans of the lake were used to determine the measurement points [40]. Field measurements were performed once a month from June 2019 to May 2020 at eight points on the reservoir and one on the river near the mouth of the lake (Figure 3). During the field tests, the value and direction of water movement, as well as the direction and speed of the wind were measured and the physicochemical parameters of water were measured. The pontoon used to move between the measurement points was restrained, with each measurement using two anchors for better stabilization. The JDC Electronic Skywatch Pro anemometer was used to measure the wind speed. The wind direction was measured with a windlass and compass on the Garmin Oregon 650 t GPS receiver. Wind speeds were measured in km/h at about 1 m above the water surface. The Hanna HI 9828 multiparameter gauge was used for physicochemical measurements–water temperature and electrolytic conductivity. Measurements were made at the surface and at the bottom. Samples for chemical analysis were taken at each point. A peristaltic pump was used for sampling. Until the analysis in the chemical laboratory, the frozen samples were stored in 15 mL Falcon tubes. All samples were filtered through GF/C filters (Whatman, UK). In the case of elemental analysis, the samples were additionally acidified with $HNO_3$ (Sigma-Aldrich, Darmstadt, Germany) for preservation.

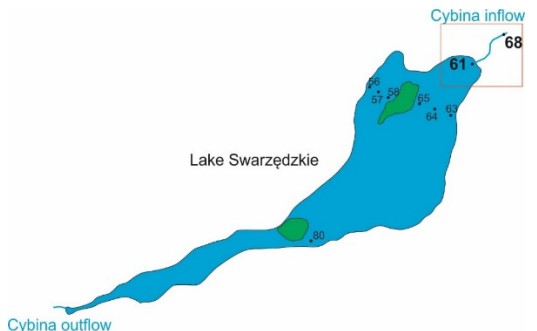
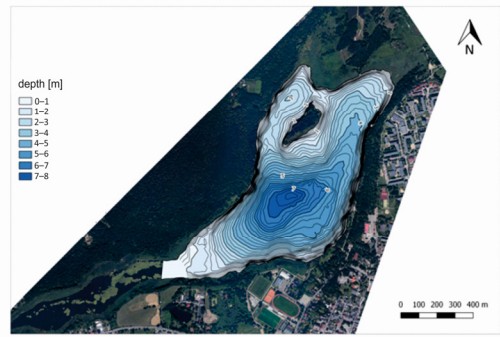

**Figure 3.** Measurement points located on Lake Swarzędzkie (**left**)-the red rectangle indicates the study area, the blue colour represents the water surface, the green the islands; bathymetric plan of the lake [40] (**right**).

The total concentration of elements (Ba, Ca, Fe, K, Mg, Mn, Na, and Zn) was determined using inductively coupled plasma optical emission spectrometry (ICP-OES). The

following conditions were repeated after Rzymski [44]. An Agilent 5110 ICP-OES spectrometer (Agilent, Santa Clara, CA, USA) was used. Simultaneous axial and radial views of the plasma were obtained by Synchronous Vertical Dual View (SVDV) using dichroic spectral fusion technology. The height for radial plasma observations was 8 mm. The RF power was 1.2 kW. The fixed Echelle classifying optics were thermostatted at 35 °C and the detector, a charge-coupled device (CCD), cooled to −40 °C using a triple Peltier system. A cyclone spray chamber, a OneNeb pneumatic nebulizer, and a quartz plasma torch (Agilent) were used. The flow rates of the nebulizer, plasma, and auxiliary argon were 0.7, 12, and 1.0 L·min$^{-1}$, respectively. The measurement time was 5 s, and one measurement consisted of three repetitions. The following analytical wavelengths (emission lines) were used: Ba II 455.403 nm, Ca I 422.673 nm, Fe II 238.204 nm, KI 766.491 nm, Mg II 279.553 nm, Mn II 257.610 nm, Na I 588.995 nm, Ni II 231.604 nm and Zn I 213.857 nm. Commercial ICP analytical standards (Romil, Cambridge, UK) were used for calibration. The detection limits (LODs, as a 3-sigma criterion) were found at the level of 1 μg·L$^{-1}$. For traceability measurements, the standard addition method was used, and the calibrated levels were within the acceptable range (80–120%). The measurement uncertainty was at the level of 5%. The concentration of anions (chlorides and sulphates) was determined by ion chromatography (IC). An 883 Basic IC Plus (Metrohm, Herisau, Switzerland) chromatograph with a conductivity detector was used. The suppressor was Metrohm Suppressor Module (MSM) with 50 mmol·L$^{-1}$ $H_2SO_4$ as regenerant. A Metrosep A SUPP 5 column, 150 × 4.0 mm, resin particle size 5 μm (Metrohm) was used. The eluent was 1.0 mmol·L$^{-1}$ sodium bicarbonate ($NaHCO_3$), 3.2 mmol·L$^{-1}$ sodium carbonate ($Na_2CO_3$) obtained from an eluent concentrate (Sigma-Aldrich, St. Louis, MO, USA). The flow rate of the eluent was isocratic, 0.8 mL·min$^{-1}$, and the injection volume was 20 μL. The above conditions were in accordance with the manufacturer's application note [45,46]. All anions eluted within 18 min. LOD was found at the level of 10 μg·L$^{-1}$.

The acoustic Doppler method was used to determine the velocity of the water currents. Measurements were made with the StreamPro by Teledyne RD Instruments. The ADCP flow meter was installed on a floating boat. The standard model was additionally equipped with a stabilizer, which improves the stabilization of the device in difficult weather conditions. The ADCP device recorded the horizontal components of velocity along the N-S and E-W axes and the direction of water movement. Measurement verticals have been divided into 25-cm sections. The thickness of the unmeasured layer, resulting from the distance of blanking and reflection of the signal from the non-uniform bottom, was the sum of 6% of the depth from the head to the bottom and the height of one measuring cell [47]. The device used has inclination and heel sensors to correct measurements during the rolling or tilting of the boat. The section by section (SxS) method with the creation of measurement verticals was used in the measurements. One measured vertical corresponded to one measuring point. Considering the dynamics of lake water and the size of the reservoir, it seems to be the most advantageous measurement method. The measurement time for each riser was 300 s. SxS Pro, and WinRiver II software from Teledyne RD Instruments was used for processing the measurement data.

The pictures were taken with a Yuneec H520 unmanned aerial vehicle (UAV) equipped with an E10T camera. The camera consists of two optical systems in which light-sensitive matrices are mounted-one in the RGB color space and the other thermal. Table 1 shows the basic operating parameters of both modules.

The camera records and saves two images simultaneously. In addition, because of the digital processing of the thermal image, an image with a resolution of 640 × 512 is created, which has a dynamic scale and is saved as a third file in the RGB color space with dynamic color settings.

The air space on Lake Swarzędzkie is divided into two aviation zones, such as the Aerodrome Trafic Zone (the zone where the Cybina River flows into Lake Swarzędzkie-thermal research area, up to 120 m with the consent of the zone manager, which the authors received) and the controlled zone of the civil airport "Ławica"-outside the research area).

The flight was performed up to 30 min before sunrise. Therefore, the UAV did not have to be equipped with additional devices, and, in accordance with Polish legislation, the flight was reported to Flight Information Services.

**Table 1.** Parameters of the E10T camera suspended under the BSP Yuneec H520 [48].

|  | **RGB** | **Thermal** |
|---|---|---|
| Resolution | 1920 × 1080 Pix | 320 × 256 Pix |
| Viewing angle (FOV) | 89.6° | 34° |
| Wave recording range | 0.45–0.77 μm | 8–14 μm |
| Sensitive | - | <50 mK, @f/1.0 |
| Recording format | JPG | TIFF |
| Color tune | RGB 24-bit | TIFF 16-bit |

The flight was planned according to the path with longitudinal and transverse coverage equal to 80%, i.e., so that each object was in 25 photos. The flight was planned at a height of 100 m, which allows for an orthophotomap with the resolution shown in Table 2.

**Table 2.** Spatial differentiation of products made in Agisoft Metashape [49].

|  | **RGB** <br> **1920 × 1080 Pix** | **Thermal RAW** <br> **320 × 256 Pix** | **Thermal Enhanced** <br> **640 × 512 Pix** |
|---|---|---|---|
| Elevation model of land cover | 7.71 cm/pix | 24.7 cm/pix | 12.9 cm/pix |
| Orthophotomap | 7.71 cm/pix | 24.7 cm/pix | 12.9 cm/pix |

## 3. Results

Lake Swarzędzkie is characterized by a high value of the electrolytic conductivity of water. The mean annual value of the lake water conductivity is 636 μS·cm$^{-1}$, and of Cybina's water is 675 μS·cm$^{-1}$. These values are lower than those recorded during the lake's research in 2012 when the conductivity values were generally in the range of 550–880 μS·cm$^{-1}$ [50]. This may be due to the sustainable reclamation of Swarzędz Lake that was carried out in 2012–2014. As a result, the values of some physicochemical parameters improved [51]. However, in the 2019 vegetation season, the values of some of them returned to the values from before the restoration, which may indicate the impermanence of the treatments carried out. In the surface layer of water, the conductivity is characterized by slight spatial differentiation across the entire reservoir. Slightly higher values were observed at point 61 near the tributary of Cybina. It is related to the mixing of lake water with the water of the watercourse, which has a significantly higher conductivity. However, due to intensive mixing and low flow values, this does not result in a significant increase in conductivity at this point (Figure 4).

The extent of mixing of river water with lake water can be determined by measuring the concentrations of elements and chemical compounds in the lake and in the river. The values of these concentrations for the elements studied are shown in Figure 5. Chloride and sulphate concentrations (Figure 5A,B) were lower in the Cybina water in relation to the water of Lake Swarzędzkie. However, the lower concentrations of these compounds in the watercourse did not translate into lower concentrations in the inflow zone to the extent that the mixing range could be determined. A similar situation was observed in the case of sodium and potassium concentrations (Figure 5C,D). The concentrations of calcium and magnesium (Figure 5E,F) were very similar in the water of Cybina and Lake Swarzędzkie. The high zinc content (Figure 5G) in point 61 did not correlate with the concentration values of this element in Cybina, where the zinc concentration was lower than in measurement point 61. The most significant differences in concentrations between the water of the stream and lake water were observed in the case of iron and manganese (Figure 5H,I). However, they did not translate into an increase in concentrations near the tributary (especially in

the case of manganese) due to the rapid precipitation of these elements in lake water. On the reservoir scale, the spatial variability of the barium concentration (Figure 5J) was small, but despite small differences, in the case of this element, the river's influence on the lake is noticeable. Cybina's water had higher barium concentrations as compared to lake water, and the content of this element at measurement point 61 was higher than in the rest of the reservoir.

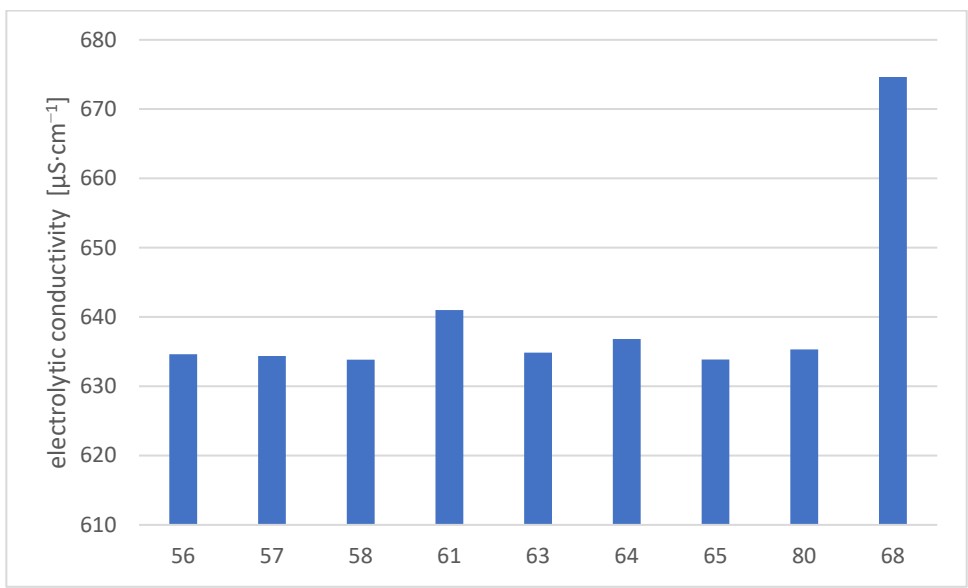

**Figure 4.** Variation of the annual average conductivity in measurement points in the water of Lake Swarzędzkie and Cybina river.

The analysis of the directions and velocities of water flow in these places showed that in terms of dynamics, the movements of water at point 61 near the tributaries are characteristic of lake water. Only in March and January, when theoretically the flows in Cybina are the highest, the directions of water movement at the bottom were observed in line with the river flow direction (SW). Currents at point 61 near the seabed, with a direction like that of the river (S and W), have already been observed for most measurements. This may mean that the waters were mixed in the bottom zone, but considering the current velocity values, which do not differ from those in the rest of the reservoir, this cannot be clearly confirmed. Table 3 summarizes the wind directions and the directions of the currents at the surface and at the bottom at point 61 during the measurement period.

Swarzędzkie Lake, according to the thermal classification of Polish lakes proposed by Sobolewski [52] is a metaepitermic reservoir. The temperature of the water surface layer shows a sinusoidal seasonal variation, typical for lakes in the temperate zone. The highest temperatures in the research period were recorded in June 2019, and the lowest in January 2020. The temperature at the measurement points did not vary vertically, which confirms the polymictic nature of the reservoir. The exception is the measurement from June 2019. At the deepest measurement points, the temperature difference between the surface and the bottom was then almost 10 °C. The spatial differentiation of temperatures across the lake was very small (Table 4). The temperature at measuring point 61 was generally slightly lower than the rest of the reservoir, on average by 0.3 °C. Mostly it was related to the lower temperature of Cybina water compared to the temperature of lake water. It was not a rule, however, because the lower temperature at point 61 was not always associated with the lower temperature of Cybina's water. These differences may also result from the non-simultaneous measurement of the temperature at the tested points and, therefore, the different degrees of heating of the water surface by the sun's rays.

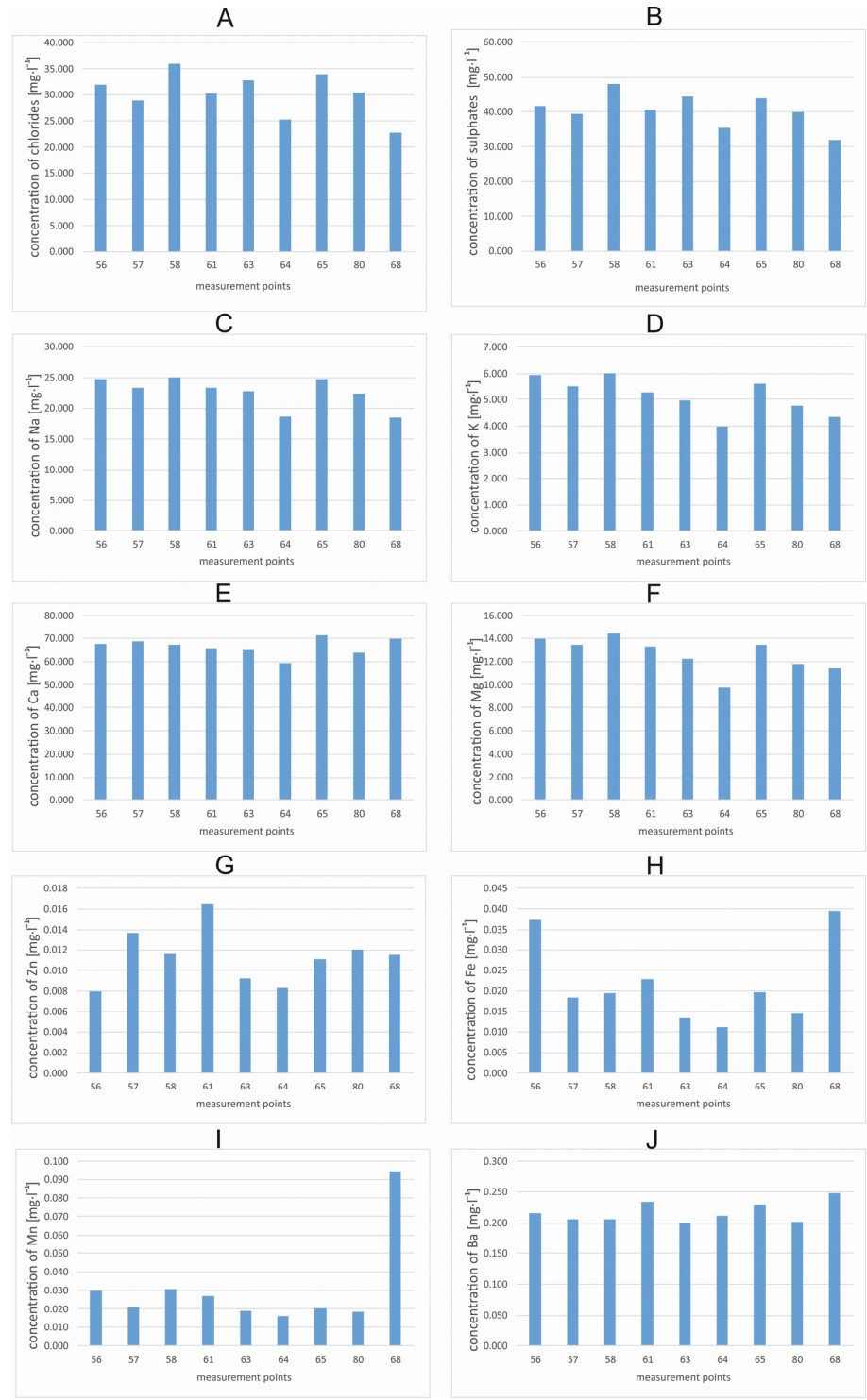

**Figure 5.** Variation of the annual average Concentration of chemical elements and compounds in measurement points in the water of Lake Swarzędzkie and Cybina river. (**A**) chlorides, (**B**) sulphates, (**C**) sodium, (**D**) potassium, (**E**) calcium, (**F**) magnesium, (**G**) zinc, (**H**) iron, (**I**) manganese, (**J**) barium.

Thermal images taken from a height of about 100 m above the water surface allowed to determine the temperature differences between the water of Cybina and the water of Lake Swarzędzkie. 141 vertical photos were taken from a height of 100 m above the starting point, which was located near the test site. Figure 6 shows the place where individual photos were taken (blue) with the optical axis (black), the starting point (red point), outline of the reservoir (yellow line) and the generated 3D model.

**Table 3.** Wind directions and speeds, and surface and bottom currents directions measured at point 61 during the field survey.

| Date of Measurement | Wind Direction | Wind Speed [km/h] | Current Direction near the Surface | Current Direction near the Bottom |
|---|---|---|---|---|
| 17 June 2019 | NW | 22 | E | W |
| 9 July 2019 | W | 16 | NE | W |
| 22 August 2019 | N | 0 | N | N |
| 26 September 2019 | SW | 5 | NE | N |
| 16 October 2019 | S | 6 | NE | S |
| 22 November 2019 | SE | 3 | N | W |
| 15 January 2020 | S | 22 | NW | SW |
| 6 February 2020 | W | 15 | NW | W |
| 27 March 2020 | N | 4 | W | SW |
| 22 May 2020 | SE | 6 | N | W |

**Table 4.** Water temperature [°C] in Swarzędzkie Lake and Cybina at the surface and at the bottom at all measuring points.

| | | 56 | 57 | 58 | 61 | 63 | 64 | 65 | 80 | 68 |
|---|---|---|---|---|---|---|---|---|---|---|
| VI 2019 | bottom | | 23.96 | | 24.58 | 17.43 | 15.64 | 24.96 | 24.87 | 23.75 |
| | surface | | 25.76 | | 24.93 | 25.43 | 25.17 | 25.23 | 27.79 | 23.75 |
| VII 2019 | bottom | 18.92 | 18.94 | 19.16 | 18.73 | 18.99 | 18.98 | 19.06 | 17.47 | 16.49 |
| | surface | 19.18 | 19.15 | 19.36 | 18.88 | 19.02 | 19.03 | 19.08 | 18.04 | 16.49 |
| VIII 2019 | bottom | 21.27 | 21.56 | 21.72 | 21.06 | 21.24 | 21 | 21.65 | 20.38 | |
| | surface | 22.89 | 22.87 | 22.58 | 21.4 | 21.89 | 21.83 | 22.21 | 23.08 | |
| IX 2019 | bottom | 15.31 | 15.24 | 15.33 | 15.38 | 15.13 | 15.06 | 15.24 | 14.48 | |
| | surface | 15.38 | 15.34 | 15.35 | 15.39 | 15.34 | 15.37 | 15.37 | 15.21 | |
| X 2019 | bottom | 13.49 | 13.45 | 13.49 | 13.38 | 13.4 | 12.69 | 13.33 | 12.98 | 10.8 |
| | surface | 13.51 | 13.45 | 13.49 | 13.39 | 13.43 | 13.38 | 13.34 | 12.98 | 10.8 |
| XI 2020 | bottom | 7.26 | 7.22 | 7.23 | 7.02 | 7.15 | 7.16 | 7.18 | 7.46 | 7.14 |
| | surface | 7.29 | 7.28 | 7.24 | 7.11 | 7.16 | 7.17 | 7.18 | 7.46 | 7.14 |
| I 2020 | bottom | 3.02 | 3.05 | 3.02 | 2.77 | 2.86 | 2.89 | 2.96 | 3.19 | 3.6 |
| | surface | 3.04 | 3.07 | 3.02 | 2.88 | 2.87 | 2.89 | 2.96 | 3.19 | 3.6 |
| II 2020 | bottom | 3.45 | 3.49 | 3.56 | 3.29 | 3.49 | 3.51 | 3.46 | 3.68 | 3.41 |
| | surface | 3.49 | 3.51 | 3.56 | 3.32 | 3.5 | 3.51 | 3.47 | 3.7 | 3.41 |
| III 2020 | bottom | 6.57 | 6.01 | 5.95 | 6.11 | 6.36 | 5.89 | 6.8 | 7.18 | 6.9 |
| | surface | 7.01 | 7.08 | 6.25 | 6.58 | 6.81 | 6.94 | 6.93 | 7.42 | 6.9 |
| V 2020 | bottom | 15.64 | 15.46 | 15.63 | 15.12 | 15.69 | 14.99 | 16.07 | 16.74 | 15.98 |
| | surface | 16.2 | 16.18 | 15.77 | 15.97 | 16.15 | 16.15 | 16.12 | 16.96 | 15.98 |

During the flight, the air temperature was −1.4 °C near the ground. The raid was carried out on 25 April 2021 at 3:54–4:01 UTC (+ 2 h local time) during sunrise.

Figure 7 shows the temperature changes in the water mixing zone. During the measurement, the water temperature in the stream was about 1.5 °C higher than the water temperature in the reservoir. Even before it entered the lake, the temperature of Cybina's water was dropping. The thermal image shows that the river spills in this place, creating many small channels and seeping through dense reeds towards the reservoir. Thus, temperature equalization took place before the river water entered the lake. Clear, rapid drops in temperature near 50 and 100 m from the beginning of the coordinate system in

the diagram (Figure 7) result from the fact that the thermal image of the water is obscured by vegetation. The Cybina River is about 1.5 m wide, and the pixel size is about 25 cm, which means that the vegetation on the riverbank and the bottom interfere with the measurement of the water temperature. Pixel is an average value observed on a given unit area [20,21]. The vegetation had a lower temperature, close to the air temperature, and by averaging the electromagnetic radiation of water and plants, lower readings on the thermal orthophotomap were obtained.

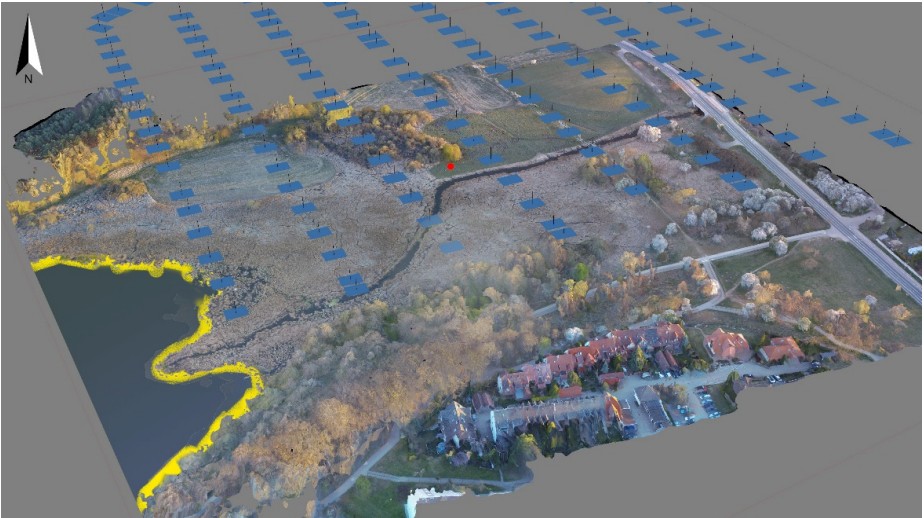

**Figure 6.** Location of photos (blue rectangles) over the three-dimensional model of the research area, outline of the reservoir (yellow line) and starting point (red point).

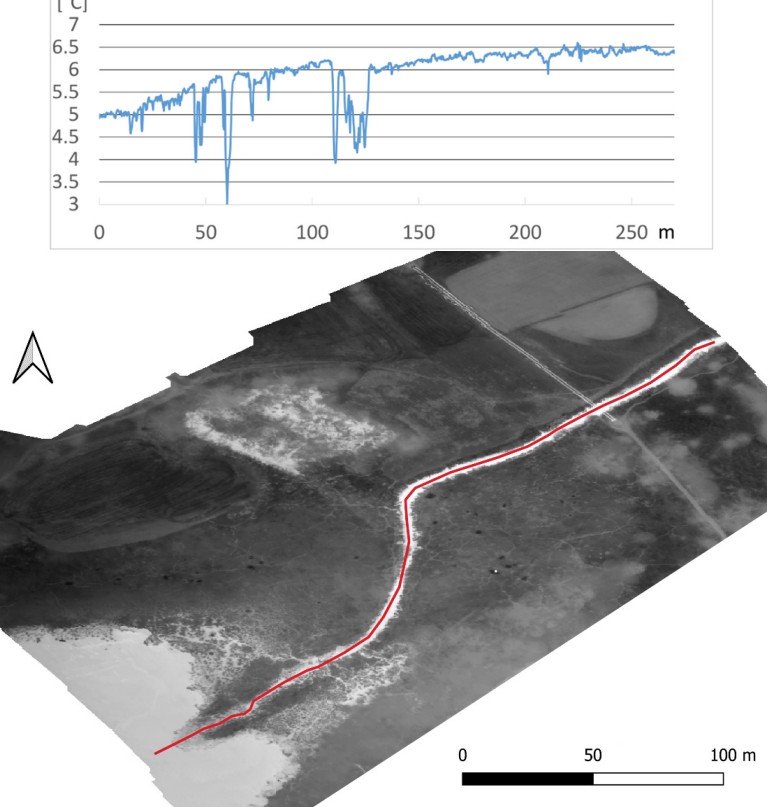

**Figure 7.** Temperature changes at the mouth of Cybina to Lake Swarzędzkie. The red line represents the path in which the temperature measurements were taken, shown in the graph with the blue line.

## 4. Discussion

Determining exactly how river and lake waters mix in the estuarine zone is complicated, especially for a river like the Cybina, characterised by low flows. Most of the available literature deals with the mixing of the waters of large rivers with those of extensive lakes, reservoirs, or sea bays [4,5,7,53–56]. Geddes [57] describes the influence of the large Murray River on the small and shallow Lake Alexandrina. However, the stratification and depth of the lake have a significant impact on the time taken for particles to move through the lake; this time can be several months at the deepest sites and a few days at the shallowest sites [9]. However, there are still few literature items on the mixing of waters of such a small river as the Cybina with waters of shallow polymictic lakes such as Lake Swarzędzkie. Determination of the extent of river mixing by measurement of electrolytic conductivity may give valuable results if the river and lake waters differ strongly in the value of electrolytic conductivity, especially if a device with automatic temperature compensation is available. However, in this case, a large number of measurements are needed to obtain a good resolution of the test, which can significantly affect their time consumption. In addition, the conductivity value can also change due to the absorption of mineral substances by the developing biomass in the lake waters. River and lake waters often differ in the concentration of different chemical elements. The quantitative composition and proportions of individual elements depend on the chemical composition of the soils and rocks of the catchment area. The origin of substances found in surface water also varies. They can be substances naturally occurring in water and those that have been introduced into it as a result of human activity [58]. Knowing the concentration of elements and chemical compounds in lake and river waters is essential for assessing water quality and helpful in understanding how ecosystems function [59]. Determination of the extent of mixing of river and lake waters in the estuary zone based on measurement of elemental concentrations requires knowledge of the processes occurring in natural waters. The complexity of these processes means that determining the extent of mixing of waters by this method is rarely used. Methods using easier measurement of parameters such as electrical conductivity, pH, dissolved oxygen, and temperature are more commonly used [60,61]. Currently, the isotope composition method is increasingly being used to determine the intensity of river-lake mixing and inflow-outflow transect; this is especially applicable to large river-lake systems [7]. Not all the elements tested showed definite differences in concentrations in the river and lake. This is the case for chloride, sulphate, sodium, potassium, calcium, and magnesium (Figure 5A–F) which precludes their use in determining the extent of mixing. Iron and manganese (Figure 5H,I) have a high affinity for phosphorus and are rapidly precipitated in lake waters so that their concentrations are generally equalised within the reservoir and cannot be used to determine the extent of the river in the lake [62,63]. Zinc (Figure 5G) concentrations in water are influenced by surface runoff, release from bottom sediments, precipitation from the water column at elevated pH values, and adsorption on the organic matter [64], which also precludes its use to determine mixing reaches. Barium (Figure 5J) concentration analysis, on the other hand, may give satisfactory results indicating the way in which the waters are mixed, as barium does not precipitate as quickly as iron or manganese, but the variation in concentrations is small. This would require many water samples to be taken and tested. Acoustic devices, such as those used during the measurements in Lake Swarzędzkie, make it possible to measure even small velocities of water movement. Measurements of flow currents showed that already a few meters near the inflow of the watercourse (point 61), in terms of dynamics, water movements are characteristic of lake waters. In order to obtain more accurate data, the measuring points would have to be considerably denser and extended with measuring points on the river, which would considerably increase the time consumption of the measurements. The measurement of water temperature in situ, with an appropriate density of measurement points, can produce satisfactory results. However, the problem is the time span of the measurements, which results in a different degree of heating up of the water surface by sunrays, which adversely affects the reliability of the measurements. This disadvantage

is eliminated by the measurement of water temperature by taking thermal images from a low aerial ceiling. It makes it possible to assess the mixing of waters simultaneously for a large area. However, in this method, the temperature of only the surface layer of water is measured, which in the case of mixing of water in the bottom zone, may distort the results. Taking thermal photos from a height of 100 m makes it easier to make thermal orthophotos because of the number of characteristic points in the photos increases. These landmarks are detected by software using the Structure from Motion method [65]. To increase the accuracy of the temperature reading and to eliminate disturbances from vegetation partially covering water, the flight altitude should be reduced. Unfortunately, flights at low altitudes usually do not have the appropriate number of characteristic points and do not allow the correct arrangement of photos and taking orthophotos. The research area is in a depression from the north and south, overgrown with high vegetation, which increases the roughness of the terrain and reduces the wind speed. This makes it possible for UAVs to fly during higher winds, as the air masses moving in this zone have a lower speed.

## 5. Conclusions

Low-altitude thermal imaging appears to be an effective method of determining the mixing of waters of a low-flow river with waters of a shallow polymictic lake. They can also be a valuable complement to the results obtained with other methods and provide additional information on the functioning of the river-lake system in the estuarine zone. The presented results may be a good starting point for planning further studies of this type, especially under varying atmospheric conditions and for different values of river flow.

**Author Contributions:** Conceptualization, R.T. and A.M.; methodology, R.T., A.M. and J.P.; software, R.T., A.M. and J.P.; validation, R.T.; formal analysis, R.T. and A.M.; investigation, R.T., A.M. and J.P.; resources, R.T.; data curation, R.T., A.M. and J.P.; writing—original draft preparation, R.T. and A.M.; writing—review and editing, R.T. and A.M.; visualization, R.T. and A.M.; funding acquisition, R.T. All authors have read and agreed to the published version of the manuscript.

**Funding:** This work was created as part of the project, GEO+: high-quality doctoral study program at the Faculty of Geographical and Geological Sciences at the Adam Mickiewicz University in Poznań (nr POWR.03.02.00-00-I039/16)" co-financed by the European Union through the European Social Fund under the Operational Program Knowledge Education Development (POWER), Priority axis III Higher education for the economy and development, Actions 3.2. doctoral studies.

**Data Availability Statement:** Not applicable.

**Acknowledgments:** We are grateful to Adam Chudziński for his help during field measurements.

**Conflicts of Interest:** The funders had no role in the design of the study; in the collection, analyses, or interpretation of data; in the writing of the manuscript, or in the decision to publish the results.

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
