# Peer review of "Attempt to Combine Physicochemical Data with Thermal Remote Sensing to Determine the Extent of Water Mixing between River and Lake"

_remotesensing, doi:10.3390/rs14164020_

Round 1

Reviewer 1 Report

The paper “Mixing of Cybina River water in Lake SwarzÄ™dzkie” addressed an important and interesting topic in determination of mixing intensity between river and lake waters. As a case study, this paper tried to combine some physical parameters, chemical parameters, water dynamics and UAV thermal remote sensing to determine the water mixing intensity between the input river and the lake. However, several shortcomings of this paper hindered its further publication, and these issues must be carefully considered and improved. The issues are as follows:

    1) The title of this paper “Mixing of Cybina River water in Lake SwarzÄ™dzkie” didn’t give clearly and too much valuable message to the audiences. I would rather suggest use “Determining the water mixing intensity between river and lake using UAV thermal remote sensing: a case study of Cybina River and Lake Searzddzkie”, or such similar expressions.

2) Line 60 first appeared the terminology “intensity of mixing”, and this terminology was the core issue of this paper, did it indicated the 2-D or 3-D spatial distribution of the water mixing zone? And how to regard the transition zone? In this paper, the exact and accurate definition of “intensity of mixing” was not defined clearly.

3) The structure of this paper should be carefully revised. The authors utilized physical parameters, chemical parameters, water dynamics and UAV thermal remote sensing to determine the water mixing intensity. However, these methods were seemed parallel and didn’t have too much interconnections. In order to fit the scope of this journal, this paper is advised to pay more attention and add more details on the development of UAV thermal remote sensing method/model on the water mixing intensity determination. And the other methods (i.e., physical, chemical and water dynamics) were used to validate the thermal remote sensing determination or to further demonstrate the differences between the input river and the lake waters, from multi aspects and properties. More importantly, the authors should considered the extensibility of this paper’s method to other fields and scenarios.

4) Line 76 – Line 132. The introduction of the study area should be moved to the “Materials and Methods” section.

5) Line 76 – Line 132 demonstrated the chemical parameter differences between the river and the lake. However, the “Discussion” section didn’t give deeper discussion on this phenomenon from a water physical-chemical-biological process perspective. Why Fe, Mn and Ba had clear differences between the river and the lake waters? And why other chemical parameters did not?

6) As Line 307 indicated the low flow velocity characteristic of the River Cybina, the authors should give more comparisons between this paper and previous studies of common high river flow scenarios.

7) The format of all figures and tables of this paper must be further improved.

8) Line 48: citation on “Cimatoribus et al.”?

Line 260 – Line 261: Please revise this sentence with a more formal expression.

Author Response

Dear Reviewer

Thank you very much for your valuable comments on our article. We tried to answer all your objections and questions to improve our article. The issue of mixing of river and lake waters is very important for the functioning of such ecosystems. Determining this phenomenon is difficult for small watercourses, especially using traditional methods of measuring temperature and other physicochemical parameters. Therefore, we decided to use UAV thermal imaging for this purpose, which proved to be very helpful.

  1. We agree with the suggestion to change the title of the paper to one that would give the audiences a clear message as to the content of the article. The title " Attempt to combine physicochemical data with thermal remote sensing to determine extent of water mixing between river and lake " is more in line with the content of the article while indicating the purpose of the research conducted.
  2. The term "intensity of mixing" is perhaps not the terminology that best describes the phenomenon presented, and it is difficult to provide a clear definition. We decided to replace it with the term "extent of mixing", which we define as the spatial extent of the zone of mixing between river and lake waters, where parameters (such as temperature) differ from the rest of the reservoir. This extent is generally defined in 2D space in the lake under study, but one should bear in mind the possibility of mixing of waters in the bottom zone in the case of deeper reservoirs (then it would be necessary to define the extent of mixing in 3D space).
  3. We have extended our work on the remote sensing method. The use of this method provides great opportunities to estimate the phenomenon of water mixing in the estuarine zone. We have also tried to demonstrate that when traditional physicochemical methods are not sufficient to accurately determine the extent of water mixing, the remote sensing method is very helpful. It is not just a matter of using physicochemical methods to validate the thermal remote sensing method or vice versa. These methods can complement each other, and the remote sensing method even should be the starting point for determining the phenomenon of water mixing and planning further studies of this phenomenon.
  4. As suggested we moved the introduction of the study area to the “Materials and Methods” section.
  5. We have enriched the discussion of chemical element concentrations by attempting to take into account the biological and chemical processes occurring in natural waters sufficiently to determine the suitability of this type of method to describe the phenomenon of the extent of mixing between river and lake waters.
  6. We mention papers that deal with water mixing in the tributary zone, especially for river-lake systems characterized by high flows, e.g. [3][4][5][7][8][9][28][50][51]. However, we focused on the case of a flowing polymictic lake with a small stream flowing through it. There are far fewer studies on this type of system.
  7. We have improved the format of most figures and tables in the paper.
  8. We have added to the citation.

Reviewer 2 Report

Dear authors,

Your effort to collect data related to the river-lake interaction is remarkable. These data can be used to calibrate mathematical models of Computational Fluid Dynamics (CFD). 

According to https://en.wikipedia.org/wiki/Computational_fluid_dynamics Computational fluid dynamics (CFD) is a branch of fluid mechanics that uses numerical analysis and data structures to analyze and solve problems that involve fluid flows. Computers are used to perform the calculations required to simulate the free-stream flow of the fluid, and the interaction of the fluid (liquids and gases) with surfaces defined by boundary conditions. CFD is based on Navier-Stokes equations.

My suggestion is to develop the work by doing CFD simulations. 

Small remarks:

1.     The first two panels of the figure 5 are in Polish.

2.     Figure 6: I suggest to draw the outline of the reservoir. It is difficlut to correlate the Figures 3 and 6. Apparently, they do not have the same orientation.

Best regards

Author Response

Dear Reviewer

Thank you very much for your valuable comments on our article. We tried to answer all your objections and questions to improve our article. The issue of mixing of river and lake waters is very important for the functioning of such ecosystems. Determining this phenomenon is difficult for small watercourses, especially using traditional methods of measuring temperature and other physicochemical parameters. Therefore, we decided to use UAV thermal imaging for this purpose, which proved to be very helpful. The use of this method provides great opportunities to assess the phenomenon of mixing of waters in the estuarine zone. We are familiar with numerical models for simulating flows and water movement in reservoirs, and they have been used in some of the papers we have cited. They generally apply to large reservoirs, but we think that using them for small polymictic lakes and small streams could give valuable results. Especially as this paper is a fragment of a larger project and we have received a lot of interesting data, including reservoir water circulation, which we could use in the future when developing a numerical model.

  1. We have translated the description in Figure 5.
  2. We changed the orientation of figures 6 and 7 to the same as figure 3. We drew the outline of the reservoir has been added.

Best Regards

Reviewer 3 Report

Please, proceed to a language cross-checking by an English native speaker. It has not significant linguistic issues, but it needs some further improvement.

Please include the references below, so that to add a more global influence of your work, covering both: a) more geographic areas and b) similar case studies.

Kagalou I. & Psilovikos A., 2013. Assessment of the Typology and the Trophic Status of two Mediterranean Lake Ecosystems (NW Greece). Water Resources, Vol. 41, No 3, pp 335 – 343.

  Herb W. & Heinz S., 2020. Temperature Stratification and Mixing Dynamics in a Shallow Lake With Submersed Macrophytes. Lake and Reservoir Management, 20 (4), 296 - 308.  

Author Response

Dear reviewer

Thank you very much for your valuable comments on our article. As You suggested, we have added references to extend the global impact of our work.

Best Regards

Reviewer 4 Report

 1.      Line 131: In Fig. 2, the discharge data was from 1976 to 1980. It is too old. Could you update this average monthly discharge, and describe the effect of climate change on the discharge? Is it the same trend that maximum states are observed in March, while low flows generally last from June to November?

2.      Line 198: How to decide these nine points for field measurements performed once a month from June 2019 to May 2020 in Fig. 3? Is it suitable for the spatial distribution? Please explain more detail.

3.      Line 220: The values of the electrolytic conductivity of water in 2019-2020 were lower than those recorded during the lake's research in 2012. Why? What is the important reason?

4.      Line 251: In Fig. 5, it shows the results of concentration of chemical elements and compounds at individual measurement points. The high zinc content in point 61 did not correlate with the concentration values of this element in Cybina, where the zinc concentration was lower than in the measurement point 61. Conversely, the highest manganese content in point 68 is the most significant differences in concentrations between the water of the stream and lake water. It is necessary to explain in detail for highest zinc content in point 61 and highest manganese content in point 68. Don’t just show the results, please.

5.      Line 304: In Fig. 7, there are clear rapid drops in temperature near 50 and 100 m from the beginning of the coordinate system. Is this result mainly from the fact that the thermal image of the water is obscured by vegetation? Please describe the effect of vegetation on thermal image more detail.

6.      This study measured wind directions, speeds, and surface and bottom currents directions as listed in Table 3. Is there relationship between the wind direction or speed and the water temperature?

Author Response

Dear Reviewer

Thank you very much for your valuable comments on our article. We tried to answer all your objections and questions to improve our article.

  1. We have added a graph with flow values over the measurement period. It shows the decreasing trend of flow values over the year with the trend that maximum states are observed in spring (March) and minimum states in summer.
  2. As mentioned in the text, the results used in this paper are a fragment of a wider project concerning the study of water circulation in Swarzędzkie Lake. The basic criterion for the selection of measurement points was the morphometry and bathymetry of the reservoir. The idea was, among other things, to create measurement profiles on both sides of the island. Points at the inflow and outflow were selected so that changes in water dynamics could be determined in the context of the influence of the stream.
  3. A sustainable restoration of Swarzędzkie Lake was conducted between 2012 and 2014. As a result, the values of some physicochemical parameters improved (Rosińska et al., 2018). However, in the 2019 vegetation season, the values of some of them returned to the values from before the restoration which may indicate the impermanence of the treatments carried out.
  4. We have expanded the description somewhat and added more detail on the chemical element concentration values in the discussion of this article. We have tried to briefly explain why the chemical elements studied were not suitable for determining the extent of the river in the lake and what this is due to.
  5. The Cybina River is about 1.5 m wide. It is overgrown with vegetation and a 25 cm pixel shows average reflection from water and plants. The authors added an appropriate fragment.
  6. Wind speed influences the horizontal and vertical circulation of water lake. Wind direction is also important because of the distance of the wave run-up. At the same time, in the stream estuary zone, the direction and speed of water movement can be altered by the flow of the stream. The directions and velocities of water movement at point 61 are presented to show that they are characteristic of lake water and the flow of the river is not marked in the movement dynamics.

Best Regards

Round 2

Reviewer 1 Report

This work is valuable, and the author's modification and reponse fully comply with the comments.

Reviewer 2 Report

Just a suggestion to improve the clarity of the lines 30-32:

The presence of submerged macrophytes in the reservoir is also important and can significantly affect water fluxes throughout the lake[1]. Therefore, the flow may only reach the narrow zone of the river mouth or be along the entire length volume of the lake.